# Isoforms of the TAL1 transcription factor have different roles in hematopoiesis and cell growth

**Aveksha Sharma**[1☯], **Shani Mistriel-Zerbib**[2☯], **Rauf Ahmad Najar**[1¤], **Eden Engal**[1], **Mercedes Bentata**[1], **Nadeen Taqatqa**[1], **Sara Dahan**[1], **Klil Cohen**[1], **Shiri Jaffe-Herman**[1], **Ophir Geminder**[1], **Mai Baker**[1], **Yuval Nevo**[3], **Inbar Plaschkes**[3], **Gillian Kay**[1], **Yotam Drier**[2], **Michael Berger**[2], **Maayan Salton**[1]*

**1** Faculty of Medicine, Department of Biochemistry and Molecular Biology, The Institute for Medical Research Israel–Canada, The Hebrew University of Jerusalem, Jerusalem, Israel, **2** Faculty of Medicine, The Lautenberg Center for Immunology and Cancer Research, The Institute for Medical Research Israel–Canada, The Hebrew University of Jerusalem, Jerusalem, Israel, **3** Info-CORE, Bioinformatics Unit of the I-CORE Computation Center, The Hebrew University of Jerusalem, Jerusalem, Israel

☯ These authors contributed equally to this work.
¤ Current address: Department of Pediatrics, Lung Biology and Disease Program, University of Rochester School of Medicine and Dentistry, Rochester, New York, USA.
* maayan.salton@mail.huji.ac.il

**Data Availability Statement:** The data supporting the findings of this study are available within the article and its supplementary material. The data discussed in this publication have been deposited

## Abstract

T-cell acute lymphoblastic leukemia (T-ALL) protein 1 (TAL1) is a central transcription factor in hematopoiesis. The timing and level of TAL1 expression orchestrate the differentiation to specialized blood cells and its overexpression is a common cause of T-ALL. Here, we studied the 2 protein isoforms of TAL1, short and long, which are generated by the use of alternative promoters as well as by alternative splicing. We analyzed the expression of each isoform by deleting an enhancer or insulator, or by opening chromatin at the enhancer location. Our results show that each enhancer promotes expression from a specific TAL1 promoter. Expression from a specific promoter gives rise to a unique 5′ UTR with differential regulation of translation. Moreover, our study suggests that the enhancers regulate TAL1 exon 3 alternative splicing by inducing changes in the chromatin at the splice site, which we demonstrate is mediated by KMT2B. Furthermore, our results indicate that TAL1-short binds more strongly to TAL1 E-protein partners and functions as a stronger transcription factor than TAL1-long. Specifically TAL1-short has a unique transcription signature promoting apoptosis. Finally, when we expressed both isoforms in mice bone marrow, we found that while overexpression of both isoforms prevents lymphoid differentiation, expression of TAL1-short alone leads to hematopoietic stem cell exhaustion. Furthermore, we found that TAL1-short promoted erythropoiesis and reduced cell survival in the CML cell line K562. While TAL1 and its partners are considered promising therapeutic targets in the treatment of T-ALL, our results show that TAL1-short could act as a tumor suppressor and suggest that altering TAL1 isoform's ratio could be a preferred therapeutic approach.

in NCBI's Gene Expression Omnibus and are accessible through GEO Series accession number GSE214833 (RNA-seq) (https://www.ncbi.nlm.nih.gov/geo/query/acc.cgi?acc=GSE214833) and accession number GSE216684 (ChIP-seq) (https://www.ncbi.nlm.nih.gov/geo/query/acc.cgi?acc=GSE216684). All FCS files were uploaded to the FlowRepository (FR-FCM-Z69L).

**Funding:** This work was supported by the Israel Science Foundation (ISF 462/22), Israeli Cancer Association, Israel Cancer Research Foundation and the Alon Award from the Israeli Planning and Budgeting Committee (PBC). AS's research scholarship is partially funded by TEVA's bioInnovation program. MaB's research scholarship is partially supported by the Carole and Andrew Harper Diversity Program, as well as TEVA's bioInnovation program. MeB's research scholarship is partially funded by the Kaete Klasuner Scholarship. The funders had no role in study design, data collection and analysis, decision to publish, or preparation of the manuscript.

**Competing interests:** The authors have declared that no competing interests exist.

**Abbreviations:** bHLH, basic helix–loop–helix; ChIP-seq, chromatin immunoprecipitation followed by sequencing; CML, chronic myeloid leukemia; co-IP, co-immunoprecipitation; CPT, camptothecin; DMEM, Dulbecco's Modified Eagle's Medium; EDTA, ethylenediaminetetraacetic acid; FBS, fetal bovine serum; IP, immunoprecipitation; MuTE, mutation of the TAL1 enhancer; PEI, polyethylenimine; RLU, relative light unit; SDS, sodium dodecyl sulfate; T-ALL, T-cell acute lymphoblastic leukemia; TAL1, T-ALL protein 1; TSO, template switching oligo; TSS, transcription start site; 5-FU, 5-fluorouracil.

## Introduction

Tissue-specific programs of gene expression are driven by key transcription factors. T-cell acute lymphoblastic leukemia (T-ALL) protein 1 (TAL1, also known as SCL) is a pivotal transcription factor in hematopoiesis, directing specification of mesoderm differentiation into specialized blood cells. Specifically, TAL1 is expressed in hematopoietic stem cells and erythroid and megakaryocytic lineages, whereas it is silenced in B and T-cell lineages during lymphocyte development. In addition to hematopoiesis, TAL1 is also an obligatory factor in adult hematopoietic stem cells and for terminal maturation of select blood lineages. Knockdown of TAL1 in mice is lethal at embryonic day 9.5 (E9.5) due to the absence of yolk sac primitive erythropoiesis and myelopoiesis [1–3]. TAL1's role in very early hematopoietic development was delineated in chimeric mice harboring homozygous mutant TAL1−/− embryonic stem cells [4,5]. This complete block of hematopoiesis suggests a function in either the first differentiation steps from progenitor cells or the specification of mesodermal cells toward a blood fate. Knocking down TAL1 later in development (6 to 8 weeks of age) perturbed megakaryopoiesis and erythropoiesis with the loss of early progenitor cells in both lineages [6].

Aberrant expression of TAL1 is the most common cause of T-ALL [7]. TAL1 overexpression is reported to occur through chromosomal translocation, intrachromosomal rearrangement, or a mutation in the enhancer region [8]. TAL1 forms a dimer with E-proteins and in this way prevents T-cell differentiation promoted by E-protein dimerization [9]. Thus, silencing of TAL1 is a requirement for T-cell differentiation and its expression in this lineage promotes stem cell features and as a result T-ALL.

TAL1's critical role at specific time points during hematopoiesis suggests that its expression is highly regulated. To date, 5 promoters [10–13] and 3 enhancers have been identified for TAL1 [14–16]. TAL1 promoters were identified by their probable location relative to a transcription start site (TSS) and a specific chromatin signature associated with promoters, which can be a DHS site, H3K9ac, H3K27ac, or H3K4me3. TAL1 enhancers were identified by a similar chromatin signature minus H3K4me3; their connection to the *TAL1* gene required experimental validation of loop architecture. Each of TAL1 enhancers directs expression to a subdomain of the normal TAL1 expression pattern [17–19]. Enhancers −4 (all locations are in Kb relative to promoter 1a; Fig 1A) and +14 were shown to promote TAL1 expression in hematopoietic stem cells and enhancer +51 was shown to drive TAL1 expression in erythrocytes and megakaryocytes [17–19]. The connection of a specific enhancer to TAL1 promoter in a specific cell type is enabled by differential loops in the different cells [16]. These loops are established by the DNA binding protein CTCF, which upon dimerization creates a loop. Only when the enhancer and promoter are present in the same loop will a connection occur. Furthermore, insertion of just a few nucleotides at position −8 was shown to create an enhancer in patients with T-ALL [14]; this enhancer was named "mutation of the TAL1 enhancer" (MuTE). Not much is known about the pairing of the specific promoter–enhancer, and it is clear that this is part of the complex expression of TAL1. This suggests that different enhancers allow for expression of a specific isoform of TAL1 at different developmental stages.

TAL1 has 4 mRNA transcripts (TSS1-4) coding to the TAL1-long protein isoform and TSS5, which codes for the TAL1-short protein isoform (Fig 1A) [10–12]. Transcription beginning at TSS1-4 with exclusion of alternative exon 3 will also translate to TAL1-short protein [20] (Fig 1A). Both isoforms harbor the basic helix–loop–helix (bHLH) domain directly binding to DNA and were previously described in zebrafish [21–23], chickens [24], and humans [24,25] (Fig 1B).

Here, we aimed to understand the RNA processing regulation that gives rise to TAL1's 2 protein isoforms and their unique functions. While the isoforms have been studied before, to

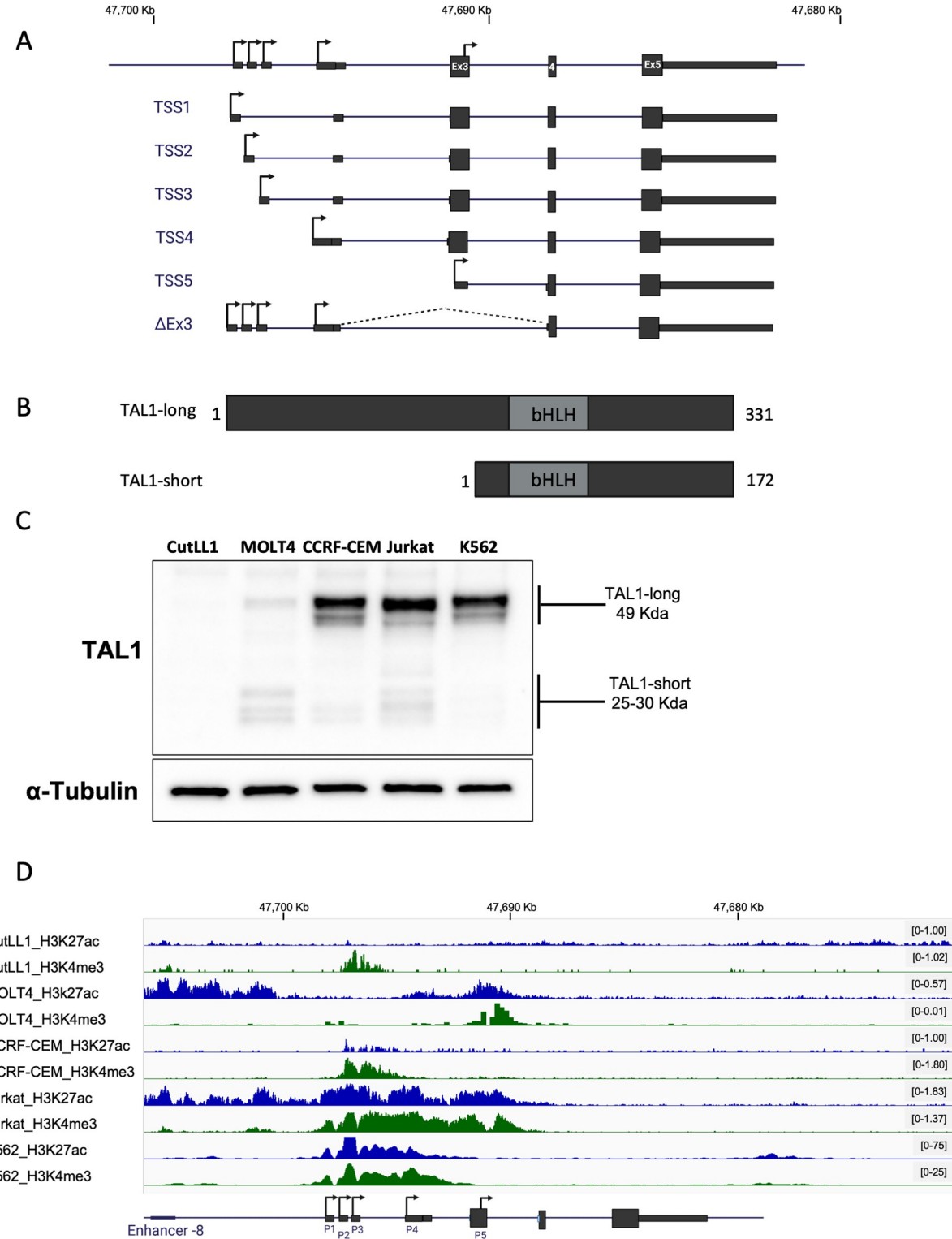

**Fig 1. TAL1 gene codes for 6 mRNA isoforms that translate to 2 protein isoforms.** (A) Schematic representation of TAL1 mRNA isoforms. Rectangles: exons, black lines: introns; arrow: transcription initiation site, coordinates corresponds to genome build hg19. (B) Schematic representation of TAL1 protein isoforms; grey rectangles represent the protein domain bHLH. (C) CutLL1, MOLT4, CCRF-CEM, Jurkat, and K562 whole cell lysate was extracted and subjected to western blot analysis using TAL1 antibody recognizing both isoforms and an endogenous control. (D) ChIP-seq tracks for H3K27ac and H3K4me3 at the TAL1 locus in the indicated cell lines (genome build hg19). Underlying data can be found in S1 Raw Images.

our knowledge, our work is the first to systemically study all aspects of their regulation and function. We manipulated TAL1's enhancers and monitored TAL1 expression as well as its alternative splicing. We took advantage of cells with activated/mutated enhancers and used a CRISPR/dCas9 system conjugated to p300 to change chromatin at TAL1's −60 enhancer. We found that the activated enhancer promotes expression as well as inclusion of TAL1 exon 3. Monitoring the chromatin at TAL1 exon 3 led us to speculate that the −60 enhancer affects exon 3 alternative splicing regulation by altering the chromatin at this exon. We further characterized the activity of the promoters and 5′ UTRs and found that while promoter 4 is the strongest, the 5′ UTR transcribed from promoter 5 is the best at promoting translation. Our findings from co-immunoprecipitation (co-IP), chromatin immunoprecipitation followed by sequencing (ChIP-seq), and RNA-seq indicate that TAL1-short is a stronger binding partner of E-proteins, stronger at binding DNA, and promotes expression of apoptotic genes. To study the role of the isoforms in hematopoiesis, we expressed each of the protein isoforms in mice bone marrow and found TAL1-short to reduce survival of hematopoietic stem cells. In addition, using the chronic myeloid leukemia (CML) cell line K562 we showed that TAL1-short promotes erythropoiesis. Finally, we demonstrated that while TAL1 was known to promote cell growth, TAL1-short expression in Jurkat and K562 cell lines stimulates cell death. In summary, our results demonstrate that TAL1 isoforms harbor differential functions in hematopoiesis and cell growth both in vitro and in vivo.

## Results

### TAL1's chromatin landscape and gene expression regulation

We set out to characterize the regulation and function of TAL1 isoforms. The prediction tool ProtParam tool [26] indicated that the half-life of the TAL1-long isoform is 7.2 h and that of the TAL1-short is 1.1 h. The differential half-life and protein length (Fig 1B) indicate the possibility that these proteins have distinct functions necessitating precise regulation. We began by evaluating the amount of the TAL1 protein and mRNA isoforms in the T-ALL cell lines Jurkat, CutLL1, MOLT4, and CCRF-CEM as well as in the CML cell line K562 (Figs 1C and S1A–S1C). In addition, we used available data to study the chromatin in inter- and intragene regions at the TAL1 gene location. To explore the regulation giving rise to TAL1 mRNA and protein isoforms, we examined H3K27ac, which indicates open chromatin at promoter and enhancer regions, and H3K4me3 marks, which indicates promoters (Figs 1D and S1C). In the CutLL1 cell line, there was no detection of TAL1 protein or mRNA and only weak H3K4me3 marks at promoter 1 to 4 regions (Figs 1C, 1D, S1A, and S1B). In the MOLT4 cell line, known to harbor the MuTE enhancer, we detected a similar amount of both protein isoforms and H3K27ac marks at an enhancer at the −8 location and promoter 5; and H3K4me3 marks at promoter 1 and 5 (Figs 1C, 1D, S1A, and S1B). The CCRF-CEM cell line harbors the SIL-TAL1 rearrangement, which allows TAL1 expression from the SIL promoter [27]. In this cell line, both TAL1 isoforms are present, with TAL1-long being dominant. Furthermore, open chromatin is only observed in the region spanning from promoter 2 to promoter 4 (Figs 1C, 1D, S1A, and S1B). As was described before, Jurkat cells overexpress TAL1 as a result of a mutation creating a strong −8 MuTE enhancer. While we could detect TAL1-short, the expression of TAL1-long protein isoform was higher (Fig 1C). Strong enhancers or super enhancers are marked by wide H3K27ac modifications, as can be seen at the −8 MuTE enhancer where the modification spreads for 15 Kb from the −8 MuTE enhancer to TAL1 exon 3 (Fig 1D). In addition, we observe broad H3K4me3 modification spanning 7.5 Kbp from TAL1 promoter 1 to promoter 5. Broad H3K4me3 modifications are correlated with strong enhancers and increased transcription [28] (Fig 1D). In the K562 cell line, H3K27ac and H3K4me3

modification marks an enhancer at +20 location and a broad modification of both histone marks at promoter 1 for 8 Kb, suggesting strong expression from TSS1-4 that is supported by the protein quantification (Figs 1C, 1D, and S1C). These results point to a unique and complex regulation giving rise to specific amounts of TAL1 isoforms.

The mRNA level of TAL1 transcripts did not match the protein amount of the isoforms (Figs 1C, S1A, and S1B). This is due in part to the real-time PCR method that was used that can only compare a specific isoform between samples and not one isoform to the other due to differential primer efficiency. In addition, while specific primers for transcription initiating from promoters 1 to 4 can be designed, such primers could not be designed for promoter 5, as its transcript has no unique sequence. Promoter 5 gives rise to the TAL1-short protein isoform as does exclusion of exon 3 (TAL1-ΔEx3). Hence, the detection of TAL1-short protein on the mRNA level is only possible by measuring TAL1-ΔEx3, which does not provide the full picture.

## Mutating TAL1 −8 MuTE enhancer and its effect on mRNA processing

To further explore the effect of known TAL1 enhancers on each isoform, we began by focusing on the effect of TAL1 −8 MuTE enhancer in Jurkat cells. We chose to use the Jurkat cell line since its MuTE enhancer is well characterized [14]. We started by performing a 5' RACE experiment in Jurkat cells to discover the active promoters in this cell line. Our results show transcription initiation from TSS1, 2, 4, and 5 (S2A Fig). For this reason, hereon, we focused on these TSSs. To study the specific connection between the −8 MuTE enhancer and TAL1 mRNA processing, we used Jurkat cells with deletion of 12 nucleotides of the −8 MuTE enhancer and insertion of exogenous TAL1-long to allow for cell growth (Jurkat Del-12) [14]. As reported previously [14], we detected a 5-fold reduction in TAL1 total mRNA amount following the impairment of the −8 MuTE enhancer (Figs 2A and S2C). Specifically, the reduction in expression was more pronounced in promoter 2 as compared to promoters 1 and 4 (S2D Fig). We cannot exclude the possibility that overexpression of TAL1-long in the Jurkat Del-12 cell line may contribute to the observed expression pattern from each promoter. Furthermore, we detected a 50% increase in TAL1-ΔEx3 in mutated −8 MuTE enhancer cells (Fig 2B). Mutation at TAL1 −8 MuTE enhancer was shown to deplete the H3K27ac from the TAL1 region [14]. We speculate that this is the reason for the reduced transcription as well as the change in alternative splicing, which is also known to be regulated by chromatin at the alternative exon [29].

## Activation of −60 enhancer and its effect on mRNA processing

We continued checking the effect of the enhancers on the isoform preference by activating the −60 enhancer in HEK293T cells by deleting a CTCF site between TAL1 promoter and −60 enhancer (S2E Fig). We selected the HEK293T cell line because it has a previously characterized enhancer and well-defined chromosome conformation. This deletion alters the genome loops, allowing for the −60 enhancer to connect with TAL1 promoters, as demonstrated by the chromosome conformation capture technique 5C, which captures frequently interacting chromatin segments [16]. Comparing parental HEK293T to HEK293T ΔCTCF, we found twice the amount of total mRNA and protein of TAL1 (Figs 2C and S2F), as seen previously [16]. While we detected more expression from promoters 1, 2, and 4 (S2G Fig), the increase in TAL1 total mRNA was accompanied by a 50% decrease in TAL1-ΔEx3 (Fig 2D). We speculate that enhancer activation led to open chromatin at the TAL1 promoter and exon 3, which mediates these changes.

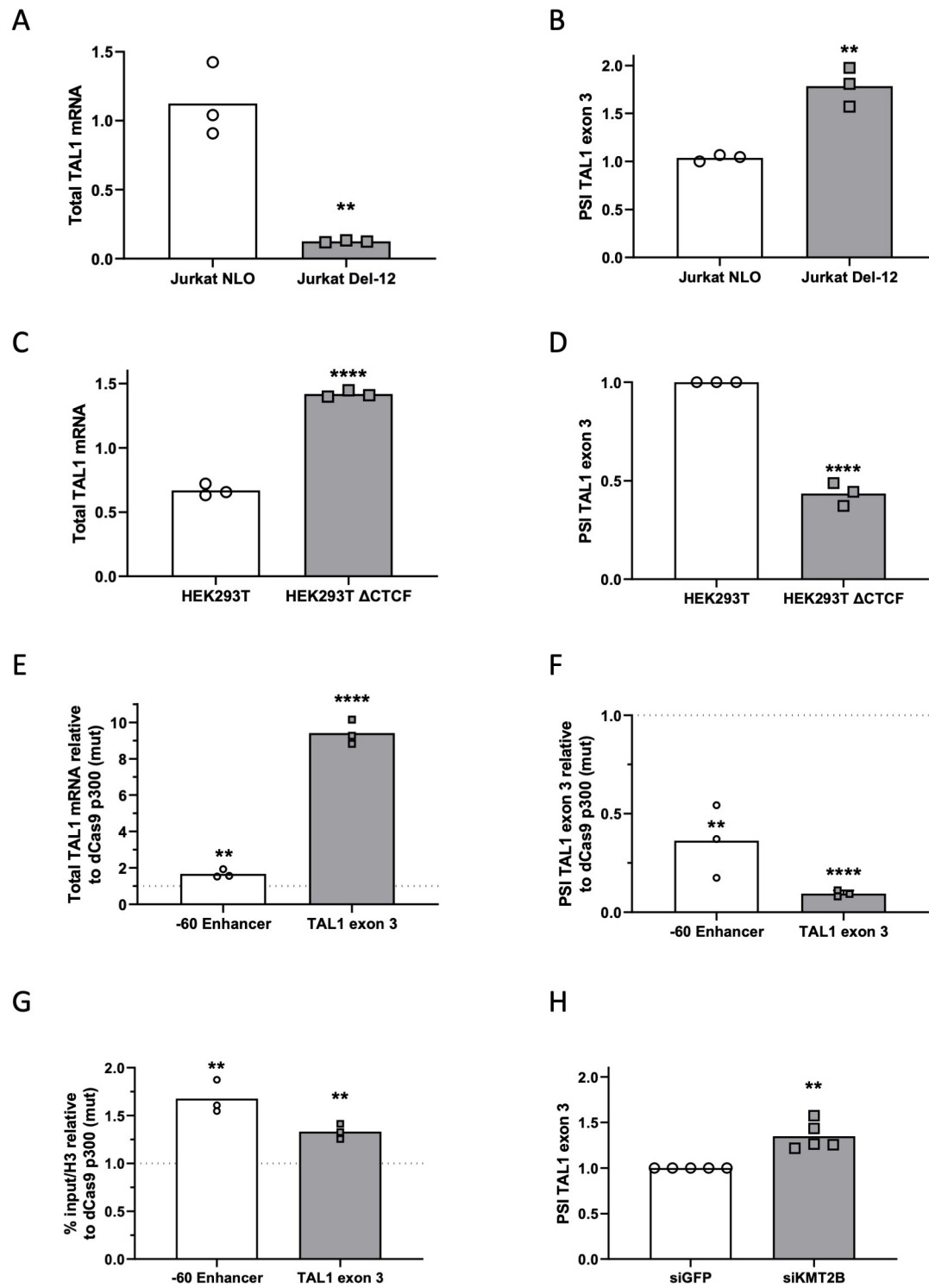

**Fig 2. TAL1 enhancers promote the expression of specific isoforms.** (A and B) RNA was extracted from Jurkat NLO cells, which express TAL1 exogenously and from enhancer mutated cells, Jurkat Del-12 and analyzed by real-time PCR for total mRNA amount of TAL1 relative to CycloA and hTBP reference genes (A) and for ΔEx3 relative to endogenous TAL1 total mRNA amount. The PSI represents the amount of mRNA molecules that include exon 3 relative to the total number of mRNA molecules that include or exclude this exon. PSI was calculated according to the formula PSI = ΔCq(inclusion isoform)/ΔCq (total mRNA) (B). Plots represent the mean of 3 independent experiments (**$P < 0.01$, Student $t$ test). (C and D) RNA was extracted from HEK293T cells and CTCF mutated cells, HEK293T ΔCTCF, and analyzed by real-time PCR for total mRNA

amount of TAL1 relative to CycloA and hTBP reference genes (C) and for ΔEx3 relative to TAL1 total mRNA amount. PSI was calculated by ΔEx3 relative to TAL1 total mRNA (D). Plots represent the mean of 3 independent experiments (****$P < 0.0001$, Student $t$ test). (E-G) HEK293T cells were transfected with either dCas9-p300 core (mut) or dCas9-p300 core (WT) with 4 gRNAs targeted to the TAL1 −60 enhancer or TAL1 exon 3 for 30 h. Total RNA was extracted and analyzed by real-time PCR for total mRNA amount of TAL1 relative to CycloA and hTBP reference genes (E) and for ΔEx3 relative to TAL1 total mRNA amount. PSI was calculated by ΔEx3 relative to TAL1 total mRNA (F). ChIP was performed of H3 pan-acetylated at the −60 enhancer and TAL1 exon 3 (G). Values are expressed as dCas9-p300 core (WT) relative to dCas9-p300 core (mut) and horizontal broken lines indicate no change between dCas9-p300 core (WT) relative to dCas9-p300 core (mut). Plots represent the mean of 3 independent experiments (**$P < 0.01$, ***$P < 0.001$, ****$P < 0.0001$ Student $t$ test). (H) Jurkat cells were transfected with siKMT2B or a negative control siRNA (siGFP), and RNA was extracted 72 h posttransfection. Real-time PCR was performed for ΔEx3 relative to TAL1 total mRNA amount. PSI was calculated by ΔEx3 relative to TAL1 total mRNA. Plots represent the mean of 5 independent experiments (**$P < 0.01$, Student $t$ test). Underlying data can be found in S1 Data.

To check our hypothesis, we manipulated the chromatin at the TAL1 −60 enhancer by tethering a general enhancer activator, the acetyltransferase p300. To this end, we used a well-characterized nuclease-deficient Cas9 (dCas9) conjugated to the p300 enzymatic core [30]. The CRISPR/dCas9 system allows us to tether a chromatin protein core domain to specific chromosome locations using a pool of 4 gRNAs. A pool of gRNAs was shown empirically to perform better than a single gRNA [30]. We selected HEK293T cells for their low expression of TAL1 compared to other cells, such as Jurkat. This feature contributes to a high dynamic range when tethering p300 core to a closed chromatin region. We cotransfected HEK293T cells with dCas9-p300 core (mut) or dCas9-p300 core (WT) and 4 gRNA plasmids targeting TAL1 −60 enhancer (S2E and S2H Fig). Our results demonstrate that tethering p300's enzymatic core to TAL1 −60 enhancer up-regulates TAL1 total mRNA one and a half-fold and gives rise to 50% less TAL1-ΔEx3 (Fig 2E and 2F) similar to activation of the −60 enhancer by deleting the CTCF site (Fig 2C and 2D). In addition, we monitored acetylation in the TAL1 region by performing ChIP with H3 pan-acetylation. Our results show that tethering p300 to TAL1 −60 enhancer alters acetylation not only at the −60 enhancer as expected, but also at promoter 5/exon 3 (Fig 2G). As we speculate that the acetylation at exon 3 could be the cause of the change in alternative splicing of this exon, we specifically acetylated promoter 5/exon 3 by tethering p300 to this region. We cotransfected HEK293T cells with dCas9-p300 core (mut) or dCas9-p300 core (WT) and 4 gRNA plasmids targeting promoter 5/exon 3, which led to 9 times more TAL1 total mRNA (Fig 2E). We speculate that the main increase in total mRNA is from transcription beginning from promoter 5. In addition, open chromatin at this region led to 80% less TAL1-ΔEx3. We hypothesize that this could be a result of reduced overall use of promoters 1, 2, and 4 and reduced exclusion of exon 3 by the open chromatin in this region.

To gain a deeper understanding of the mechanistic link between the MuTE enhancer, the broad H3K4me3 modification spanning 7.5 Kbp from TAL1 promoter 1 to promoter 5 (Fig 1D), and alternative splicing, we manipulated KMT2B, which is a component of the SET1/COMPASS complexes responsible for methylating H3K4 [31]. Specifically, we silenced KMT2B in Jurkat cells by transfecting them with siRNA targeting KMT2B or a negative control (siGFP). Our results demonstrate that silencing KMT2B (S2I Fig) led to a 30% exclusion of TAL1 exon 3 (Figs 2H and S2J). These findings provide further insights into the molecular mechanisms underlying the regulation of TAL1 alternative splicing and highlight KMT2B as a regulator of *TAL1* gene alternative splicing.

The connection between the chromatin at exon 3 and alternative splicing might be via RNAPII elongation kinetics [32–34], but when we slowed elongation of RNAPII in Jurkat cells using a low dose of camptothecin (CPT, 6 uM), expression was reduced but splicing of TAL1 was not changed (S2K–S2M Fig). Another possibility is that the promoter chromatin signature at promoter 5/exon 3 (H3K27ac and H3K4me3; Fig 1D) promotes inclusion, as was suggested by a genome-wide investigation [35]. This study indicated that exons with promoter-like

chromatin tend to be included. In addition, these exons were indicated to be in a linear proximity to the gene's active promoter, very similar to TAL1 exon 3 and promoters 1 to 4 (Fig 1A). This suggests that TAL1 enhancers affect exon 3 alternative splicing regulation by altering the chromatin at this exon.

## Characterization of TAL1 promoters and 5′ UTR

To study the characteristics of TAL1 promoters, we considered the effects of the 500 nt before each TSS. Since the 500 nt before TSS1-3 are overlapping, we cloned the 500 nt before TSS3 as representing all 3 promoters. For TSS5, we studied the sequence in exon 3 and used exon 4 sequence as a negative control. All sequences were cloned upstream to a minimal SV40 promoter in a luciferase reporter plasmid and cotransfected to HEK293T cells with renilla as an internal control. We used HEK293T cells because their transfection efficiency is higher compared to Jurkat cells. While exon 4 was similar to the minimal promoter alone, promoters 1 to 3 showed 3 times more luciferase expression, promoter 4 five times more, and promoter 5 twice the amount of luciferase expression relative to control (Fig 3A). This result suggests that promoter 4 is stronger at transcription initiation relative to promoters 1 to 3 and 5.

Since TAL1 is a transcription factor and is known to autoregulate its own transcription [9,36], we wondered whether TAL1 isoforms have an effect on either of the promoters. To this end, we cotransfected each of the promoters cloned before luciferase and renilla as internal control with either empty vector, TAL1-long or TAL1-short (S3A and S3B Fig). Our results show that TAL1-short promotes expression from TSS1-3 while TAL1-long does not (Fig 3A). This result suggests that TAL1-short has a preference for the TSS1-3 promoters.

To study the 5′ UTR characteristics generated by each of the promoters, we cloned the 5′ UTR following promoters 1, 2, 4, and 5 to the region before the luciferase initiation codon. In order to assess the mRNA stability conferred by the 5′ UTR, we measured luciferase total mRNA using real-time PCR relative to renilla as control. Our results did not detect any change in mRNA abundance between the different 5′ UTRs and an empty control (S3C Fig), suggesting that the 5′ UTRs do not affect mRNA stability. To study the 5′ UTR effect on translation,

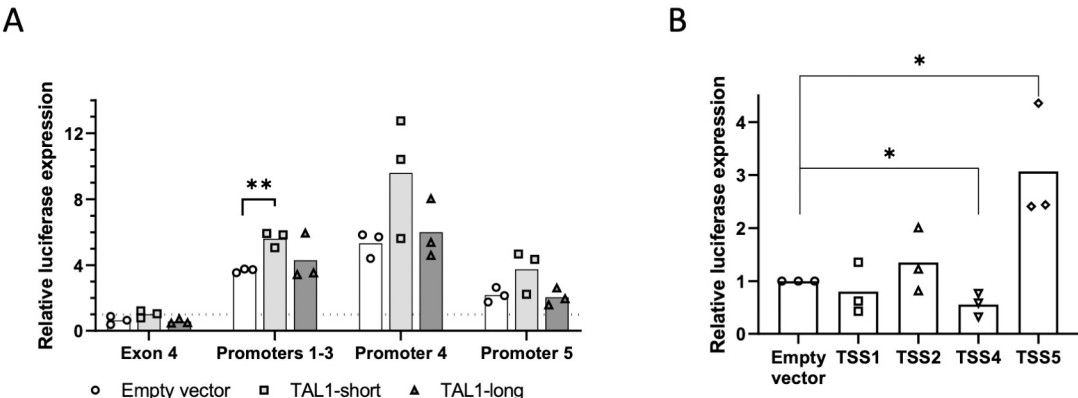

**Fig 3. TAL1-short promotes transcription from TSS1-3.** (A) HEK293T cells were cotransfected with TAL1 promoters: promoters 1–3, 4, 5, and exon 4 as a negative control. The second plasmid was an empty vector, TAL1-short or TAL1-long. After 30 h, luciferase activity was calculated relative to renilla. Horizontal broken lines indicate transfection with the empty luciferase. The mean was calculated from 3 independent biological experiments, each performed with 3 technical replicates (**$P < 0.01$, Student $t$ test). (B) HEK293T cells were cotransfected with TAL1 5′ UTRs: TSS 1, 2, 4, and 5 and empty vector as negative control. After 30 h, luciferase activity was calculated relative to renilla. The mean was calculated from 3 independent biological experiments, each performed with 3 technical replicates (*$P < 0.05$, Student $t$ test) are shown. Underlying data can be found in S1 Data.

we quantified luciferase expression relative to renilla. Our results show that the 5′ UTR generated from promoter 4 reduced luciferase expression by 30% in contrast to the 5′ UTR generated by promoter 5, which gave rise to 3 times more luciferase expression (Fig 3B). Interestingly, while promoter 4's sequence had a stronger effect on transcription, its 5′ UTR gave rise to reduced translation efficiency. This result indicates that the 4 TAL1 5′ UTRs have different characteristics, suggesting additional regulation to that of transcript level.

## TAL1-short binds more strongly than TAL1-long to E-proteins

Our investigation indicates complex mRNA processing, which we speculate is important to give rise to a precise amount of TAL1 protein isoforms at a specific point in time during hematopoiesis. To understand the importance of generating a specific amount of each isoform, we began by studying the function of the protein isoforms in Jurkat cells. TAL1 is a class II bHLH transcription factor that forms an obligate heterodimer with the class I bHLH E-proteins, which include transcription factors E2A (TCF3) and HEB (TCF12). Binding of TAL1 to E-proteins was shown to inhibit T-cell differentiation and to promote T-ALL by creating an oncogenic transcription signature [37,38]. To study the function of each of the protein isoforms, we transfected Jurkat cells with either TAL1-short or TAL1-long with a FLAG or GFP tag. We immunoprecipitated each isoform using the tag and checked for TAL1 core complex protein partners. Our results show that TAL1-short immunoprecipitated more of TCF3 and TCF12 relative to TAL1-long (Figs 4A and S4A). Binding of TAL1 to E-proteins prevents T-cell differentiation, which is promoted by dimerization of the E-proteins, thus differential binding of TAL1 isoforms to E-proteins could suggest a differential function during hematopoiesis.

## TAL1-short is enriched in known TAL1 binding sites

TAL1 is a transcription factor and is known to bind DNA and regulate gene expression with its coregulators. To check whether each isoform has unique DNA binding characteristics, we conducted ChIP-seq for each of the isoforms. All TAL1 ChIP-seq experiments to date used an antibody recognizing both isoforms [36,39]. To specifically assess the binding of each of the isoforms, we transfected Jurkat cells with either TAL1-short or TAL1-long with a FLAG tag and conducted ChIP-seq with anti-FLAG antibody (S4B Fig). We first examined specific known TAL1 target genes, including TRIB2, NFKB1, and NKX3.1, and detected TAL1-short binding at sites in the regulatory regions of each gene similar to previous results (S4C Fig) [39]. Next, we looked genome wide at published ChIP-seq data with an antibody against both isoforms [39] (TAL1-total) and compared it to the binding pattern of each of the isoforms. Our results indicate that binding of TAL1-short is more similar to that of TAL1-total than is TAL1-long at these locations (Fig 4B). The difference between the binding profiles of the isoforms might arise from stronger binding of TAL1-short, and from TAL1-short having more binding sites. This is also manifested in the fact that when considering all TAL1-total peaks, the level of enrichment of TAL1-short binding is 1.5-fold higher than TAL1-long (Figs 4C and S4D).

The relative distribution of TAL1-total bound regions are mainly located in gene bodies and intergenic regions of known protein-coding genes, consistent with the location of enhancer elements (S4E Fig). This distribution was similar to that of TAL1-short, but in TAL1-long, we saw more binding at promoter regions and less at gene bodies (S4E Fig).

We next sought to identify DNA motifs that were overrepresented within 200 bp of the peak of TAL1 binding in TAL1-total compared to each isoform. We identified 5 transcription factor binding motifs to be enriched in TAL1-total and shared between one or the 2 isoforms. Four of the transcription factors were previously described to be enriched in TAL1-bound

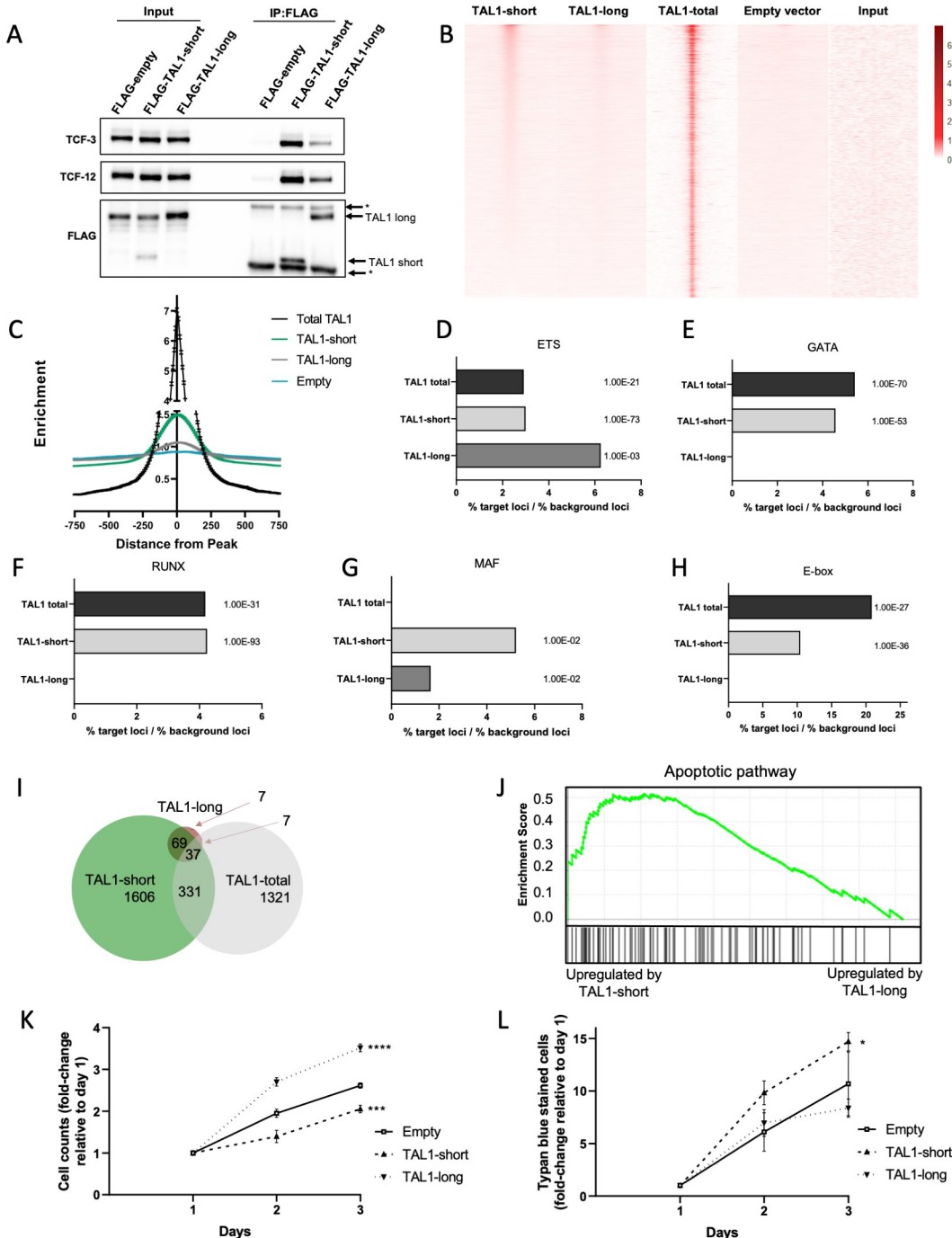

**Fig 4. TAL1-short promotes transcription of apoptotic genes and is a stronger transcription factor compared to TAL1-long.** (A-H) Jurkat cells were infected with empty vector, FLAG-TAL1-short or FLAG-TAL1-long. FLAG was immunoprecipitated and known TAL1 interacting proteins were detected with indicated antibodies (A). (B-H) FLAG-TAL1-short, TAL1-long, and empty vector's chromatin were immunoprecipitated using FLAG antibody. In addition, TAL1-total ChIP-seq data were analyzed (see methods). Heatmap showing signal enrichment over 5,876 TAL1-total binding sites (±1,000 kb from the center). Heatmap is sorted by strength of TAL1-short signal centered on ChIP-seq summits. A color scale indicating the relative signal intensity plotted on each heatmap is shown (B). ChIP-seq average signal for TAL1-total, TAL1-short, and TAL1-long and empty vector as a function of distance from TAL1-total peaks (C). Most abundant DNA sequence motifs were identified in TAL1-total, TAL1-short, and TAL1-long ChIP-seq peaks. Percentage of loci was calculated by dividing % of target sequences with motifs by % of background sequences with the motif for ETS (D), GATA (E), RUNX (F), MAF (G), and E-box (H). (I-L) Jurkat cells were infected with inducible shRNA against the 3′ UTR of TAL1. In addition, the cells were infected with either empty vector, TAL1-short or TAL1-long. Following induction with tetracycline for 72 h,

RNA was extracted and subjected to sequencing. In addition, analysis was performed on available data of TAL1-total (see methods). Venn diagram representing the overlap of TAL1-short, TAL1-long, and TAL-total target genes (I). Gene Set Enrichment Analysis (GSEA) for apoptotic pathway for TAL1-short and TAL1-long (J). Cells were seeded and counted every day following labeling by trypan blue. Live cells are plotted in (K); trypan blue-labeled cells are plotted in (L). Plots represents the mean of 3 independent experiments (*$P < 0.05$, ***$P < 0.001$, and ****$P < 0.0001$ Student $t$ test). Underlying data can be found in S1 Data and S1 Raw Images.

regions: E-box (5′-CAG[CG]TG-3′), GATA (5′-AGATAA-3′), RUNX (5′-TGTGGTC-3′), and motifs recognized by the ETS family of transcription factors (5′-GGAA-3′) (Figs 4D–4H and S4F–S4J). While the ETS motif was enriched in TAL1-total as well as in both isoforms, the rest were only enriched in TAL1-total and TAL1-short (Figs 4D–4H and S4F–S4J). In addition, we identified a new motif recognized by the MAF family of transcription factors (5′ TGCTGACT-CAGCA-3′) known to be expressed in T-ALL and to have a role in TAL1 expression (Figs 4G and S4I). This motif was not enriched in TAL1-total but was enriched in both isoforms. We speculate that the enrichment of MAF motif in both isoforms and not in TAL1-total could be due to the independent conditions of these experiments [39]. To summarize, while our results agree with previously reported TAL1 binding sites, TAL1-short exhibits stronger binding at multiple binding sites relative to TAL1-long. This result taken together with TAL1-short binding more strongly to E-proteins (Figs 4A and S4A) suggest stronger transcription properties of TAL1-short relative to TAL1-long.

## Differential targets for TAL1 isoforms

To explore the changes in expression of TAL1 targets by each of the isoforms, we conducted RNA-seq in isoform-expressing cells. First, we silenced endogenous TAL1 by stably transfecting Jurkat cells with inducible TAL1 shRNA targeting its 3′ UTR, and so targeting only the endogenous mRNA. Following treatment with tetracycline, we observed 50% silencing of total TAL1 mRNA (S4K Fig). We then transfected empty vector, TAL1-short, or TAL1-long and monitored total mRNA amount of the targets. In addition, we analyzed RNA-seq results of Jurkat cells silenced for TAL1 using shRNA [40].

Our results identify a similar number of targets for TAL1-short (2,043) as previously identified for both isoforms (TAL1-total, 1,696), while only 120 targets were attributed to TAL1-long (Fig 4I). Previous work has shown a small overlap (14%) in genes bound by TAL1 and its transcription targets [39–43] (S4L Fig). We identified a similar overlap (17%) in our results comparing results of TAL1-short ChIP-seq to RNA-seq (S4M Fig). No overlap was identified for TAL1-long, presumably due to small number of peaks and targets. We chose 3 overlapping targets of TAL1-short for validation using real-time PCR (S4N Fig). These results strengthen our hypothesis that TAL1-short is a more potent transcription factor.

TAL1-total targets were previously shown to have a role in T-cell activation and proliferation [9,39,41]. Indeed, we detect enrichment of these gene sets using the Enrichr tool [44], but only for TAL1-long and not for TAL-short (S4O–S4Q Fig). Interestingly, TAL1-short targets were enriched for genes involved in apoptosis (Figs 4J and S4R), suggesting differential cellular functions for each of the isoforms.

## Slower growth in TAL1-short overexpressing Jurkat cells

Next, we asked whether this differential expression of apoptotic genes by the isoforms will result in differential cell growth as suggested by research demonstrating that high amounts of TAL1-short in glioblastoma reduced growth [45]. To check the effect of the isoforms on Jurkat cell growth, we silenced endogenous TAL1 by stably infecting the cells with inducible TAL1

shRNA targeting its 3′ UTR. We continued by infecting FLAG-TAL1-short followed by IRES expressed GFP or FLAG-TAL1-long/GFP or empty vector. Our results show faster growth of TAL1-long overexpressing cells and slower growth of TAL1-short cells relative to empty vector (Fig 4K). Furthermore, we counted 30% more dead cells in TAL1-short expressing cells relative to empty vector (Fig 4L). These results could suggest that promoting the expression of apoptotic genes by TAL1-short and not TAL1-long hinders cell growth.

## TAL1 isoforms' role in hematopoiesis

To define the outcome of transcription activity of the 2 TAL1 isoforms, we tested their influence on hematopoiesis in a mixed bone marrow chimera setting. Enriched HSCs were transduced with retroviruses expressing either FLAG-TAL1-long/dtTomato or FLAG-TAL1-short/GFP. Then, both cells were transplanted together in lethally irradiated syngeneic recipient mice (Fig 5A). Composition of bone marrow and spleen cells were tested by flow cytometry 13 weeks after bone marrow transplantation. In comparison to WT (GFP⁻, dtTomato⁻), both TAL1-long and TAL1-short-expressing spleen and bone marrow cells showed decreased percentages of B and T cells (Figs 5B, 5C, S5F, and S5G) and a corresponding increased percentage of CD11b⁺ cells (Figs 5D, 5E, S5H, and S5I), as was previously described [46]. Similar composition of the CD11b⁺ subsets, evaluated by Ly6G and Ly6C staining, was observed in WT, TAL1-long, and TAL1-short cells (S5A and S5B Fig). In addition, focusing on the differentiation of red blood cells lineage in the bone marrow, we could observe significant decreased and increased percentages in the S0 and S3 stages, respectively, in both TAL1 isoforms expressing cells in comparison to WT cells (S5C–S5E Fig). Overall, these results demonstrate that each of the isoforms is functional in roles of TAL1 in early hematopoiesis described as supporting the myeloid over the lymphoid lineage and enhancing red blood cell maturation [46].

## Differential role for TAL1 isoforms in hematopoiesis

Having demonstrated that both TAL1 isoforms are functional in hematopoietic stem cell differentiation, we moved on to ask if they possess differential roles. To this end, we evaluated the persistency of the TAL1-expressing cells over time. Blood samples from the mixed bone marrow chimera mice were collected at different time points, and the percentages of WT, TAL1-long, or TAL1-short cells were measured by flow cytometry (Fig 5F and 5G). Three weeks after bone marrow transplantation, the recipient mice had more TAL1-short than TAL1-long expressing leukocytes (Fig 5F and 5G). However, while the percentages of the TAL1-long expressing leukocytes remained constant, those of the TAL1-short expressing leukocytes were significantly decreased over time (Fig 5F and 5G). These results suggest that TAL1-short and TAL1-long have different effects on hematopoiesis by which TAL1-short but not TAL1-long leads to hematopoietic stem cells exhaustion.

## TAL1-short promotes differentiation into erythroid cells

TAL1-short has been shown previously to favor the erythroid lineage in chicken and human cells [24]. Our results also support a unique role for TAL1-short in hematopoiesis. To further explore the role of the isoforms, we used K562 cells, which are capable of undergoing erythroid differentiation in vitro when cultured with various inducers, such as hemin. Since K562 cells express high amounts of TAL1 (Fig 1C), we began by silencing endogenous TAL1 by stably infecting the cells with inducible TAL1 shRNA targeting its 3′ UTR. Following treatment with tetracycline, we monitored 50% silencing of total TAL1 mRNA and protein (S6A and S6B Fig). We continued by infecting FLAG-TAL1-short followed by IRES expressed GFP or FLAG-TAL1-long/GFP or empty vector and treated cells with 20 μM hemin to induce differentiation

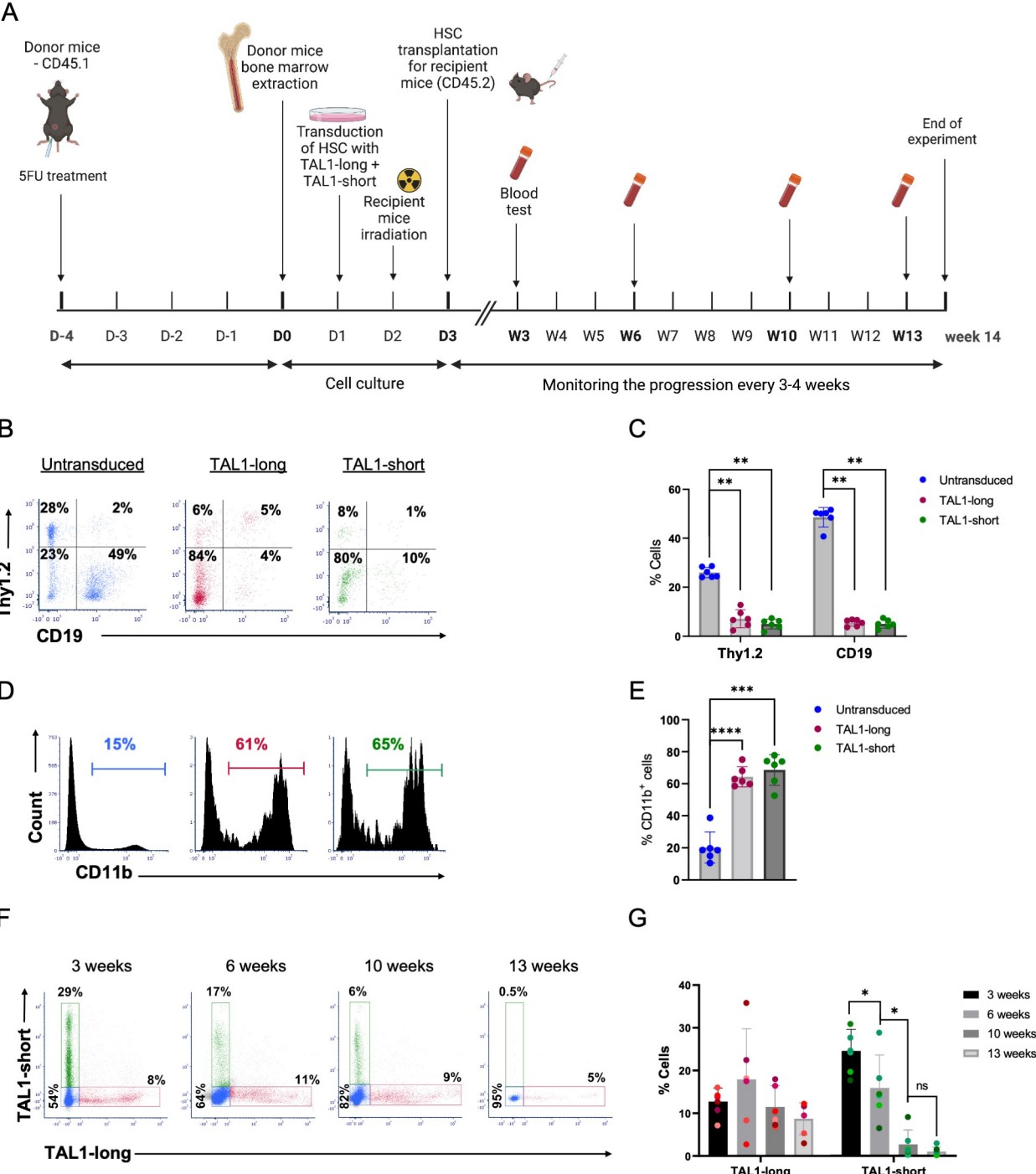

**Fig 5. TAL1-short but not TAL1-long leads to hematopoietic stem cell exhaustion.** (A) Schematic illustration of the mixed bone marrow chimera experiments. Equal numbers of bone marrow cells from 5FU-treated CD45.1 wild-type mice were transduced with retroviruses expressing either TAL1-short-GFP or with TAL1-long-dtTomato. A mixture of 1:1 ratio of the transduced bone marrow cells was then transplanted into lethally irradiated CD45.2 wild-type recipient mice. Blood samples were taken at different time points to monitor the progression of the bone marrow cell reconstitution using flow cytometry analysis. All mice were killed 14 weeks after BMT, and the spleen and bone marrow were harvested and analyzed. (B) Representative dot plots of Thy1.2 vs. CD19 staining in untransduced (left), dtTomato+ (middle), or GFP+ (right) splenocytes from the recipient mice (C) Bar graph summarizing results shown in (B). (D) Representative flow cytometry histograms of CD11b staining gated on GFP−/dtTomato− (untransduced, left panel), dtTomato+ (Tal1-long, middle panel), or GFP+ (Tal1-short, right panel) splenocytes. (E) Bar graph summarizing results shown in (D). (F) Representative dot plots of Tal1-short GFP vs. Tal1-long dtTomato staining in the blood of the recipient mice at different time points over the course of 13 weeks. (G) Bar graph summarizing results in (F). Each mouse is presented with a different shade. In (C), two-tailed paired *t* test, *< 0.05, **< 0.01, sd. (*n* = 6). Underlying data can be found in S1 Data.

(S6C Fig). We quantified the mRNA amount of α-, β-, and γ- globin, and SLC4A1, known to be induced during erythrocyte differentiation at various time points. Our results show elevated mRNA expression of α-, β-, and γ- globin and SLC4A1 erythroid markers in K562 cells expressing TAL1-short cells relative to empty and TAL1-long expressing cells without induction of differentiation (Figs 6A, S6D, and S6E), or with induction (Figs 6B, S6F, and S6G). TAL1-short's effect on uninduced cells indicates its role as a differentiation factor, a role not taken by TAL1-long.

## TAL1-short hamper survival capability in K562 cells

Our in vivo results suggest that TAL1-short might hamper the survival capability of hematopoietic cells (Fig 5F and 5G). Furthermore, TAL1-short overexpression in Jurkat cells led to slower growth and a higher number of dead cells (Figs 4K and 4L). To check the effect of the isoforms on hematopoietic cell survival, we silenced endogenous TAL1 by stably infecting the cells with inducible TAL1 shRNA targeting its 3′ UTR. Following treatment with tetracycline, we monitored 50% silencing of total TAL1 mRNA (S6A Fig). We continued by infecting FLAG-TAL1-short followed by IRES expressed GFP or FLAG-TAL1-long/GFP or empty vector. Our results show slower growth and twice as many dead cells in TAL1-short cells relative to TAL1-long and empty vector (Fig 6C and 6D). In addition, induction of erythropoiesis using hemin resulted in similar growth rates (S6H and S6I Fig). Since cells overexpressing TAL1-short show the same level of erythroid differentiation markers, we cannot determine the distinct effect of TAL1-short on differentiation and growth.

## Discussion

Previous investigation of TAL1 isoforms indicated their roles in regulation of translation and the generation of 4 isoforms [24]. Using 2 different sets of antibodies recognizing the C-terminus of the protein, common to all reported isoforms, we found 2 isoforms (TAL1-short and TAL1-long) with posttranslational modifications that affected the migration rate of the protein (Fig 1C). In contrast to a previous study [24], our results suggest regulation on the mRNA level as the initiator of the 2 isoforms; however, we found that TAL1-short promotes erythroid lineage. In addition to the results in human cell lines, work in zebrafish and chicken indicated that TAL1-short is required for erythroid differentiation, while TAL1-long is required for megakaryocytic differentiation [22].

Our ChIP-seq results suggest that some of the isoforms bind sites are shared (Fig 4B). While the location of the binding is similar between the isoforms, the enrichment was 1.5 higher for TAL1-short relative to TAL1-long (Fig 4C). Although TAL1's binding motif is commonly reported as CAGNTG [47], we did not identify this motif in our analysis of the TAL1-total or TAL1-long ChIP-seq results. However, we did observe this sequence in the TAL1-short ChIP-seq results with a $p$-value of $1 \times 10^{-93}$. We speculate that this discrepancy may be due to the complex nature of transcription factors binding and the fact that the ChIP-seq results for TAL1-total were conducted independently [39]. Nevertheless, we predict that both TAL1 isoforms bind the same DNA motif, as the bHLH domain is similar in both (Fig 1B). Quantifying the amount of TAL1-short relative to TAL1-long, we noticed that while in MOLT4, the amount was equal in CCRF-CEM, Jurkat, and K562, the amount of TAL1-long was a magnitude higher (Fig 1C). In addition, using the prediction tool ProtParam shows that the expected half-life of TAL1-long is 7 times longer than that of TAL1-short [26]. Taken together, our results suggest that while TAL1-short might bind more strongly to DNA, the amount of TAL1-long in cells might compensate for its weaker binding enabling it to have a similar effect as the short isoform.

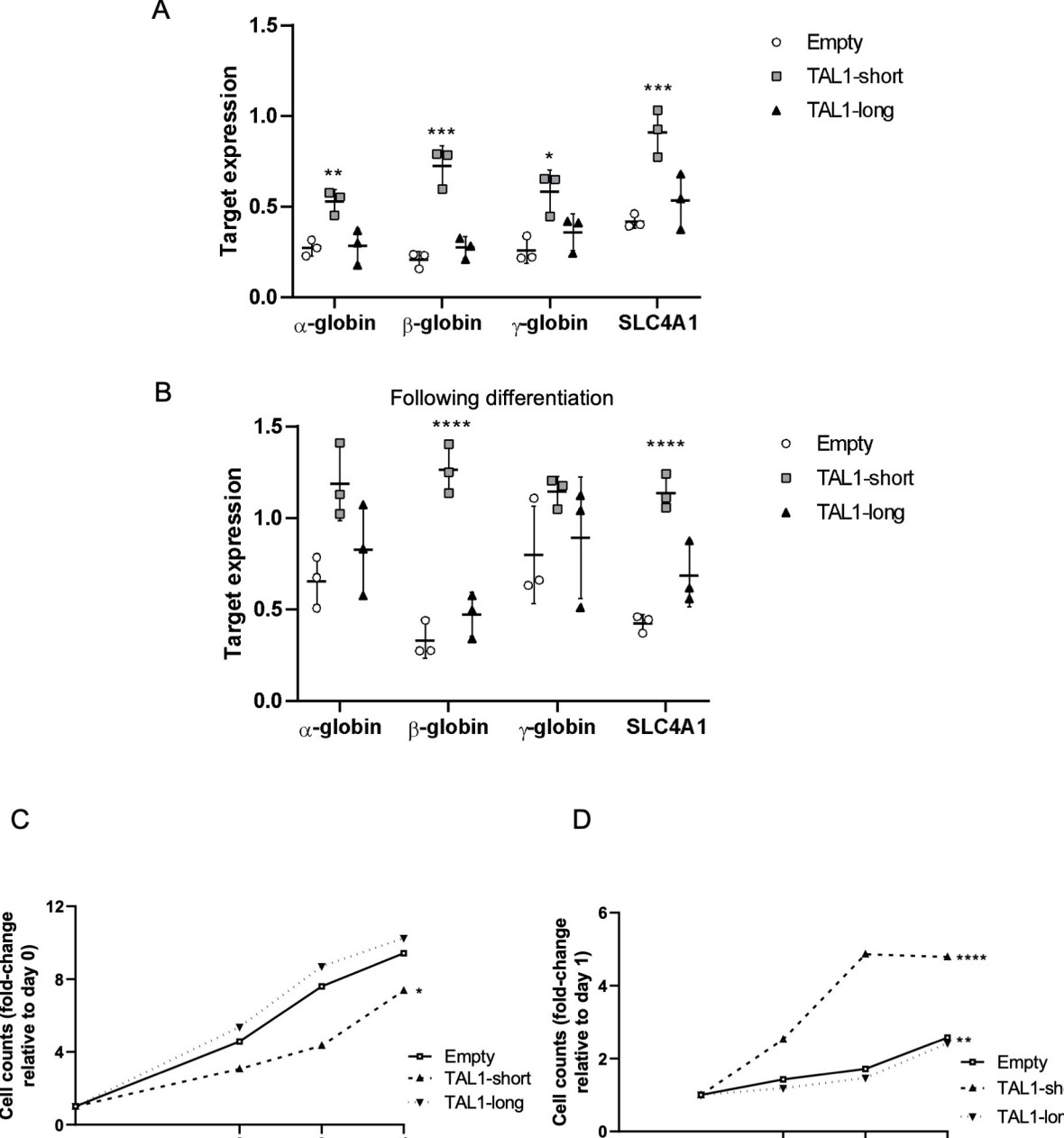

**Fig 6. TAL1-short is a stronger differentiation agent. (A-D)** K562 cells were infected with inducible shRNA against the 3′ UTR of TAL1. In addition, the cells were infected with MIGR1 plasmid with GFP at the C-terminal followed by an IRES element. Infection was with empty vector, TAL1-short, or TAL1-long. Following induction with tetracycline for 72 h, cells were treated with 20 μM hemin to promote erythroid differentiation for the indicated time points. RNA was extracted and analyzed by real-time PCR for total mRNA amount of α-, β-, and γ-globin and SLCA4 relative to CycloA and hTBP reference genes without hemin (A) and with treatment (inducing differentiation) for 48 h (B). Cells were seeded and counted every day following labeling by trypan blue. Live cells are plotted in (C) and trypan blue-labeled cells are plotted in (D). Plots represents the mean of 3 independent experiments (*$P < 0.05$, **$P < 0.01$, ***$P < 0.001$, and ****$P < 0.0001$, Student $t$ test). Underlying data can be found in S1 Data.

Our results point at TAL1-short as a stronger protein partner of E-proteins (Fig 4A). The TAL1-E-protein dimer does not operate alone and often forms a regulatory complex with other transcription factors, including LMO (LMO1 or LMO2), GATA (GATA1, GATA2 or GATA3), and LDB1 proteins. In addition, RUNX1 and the ETS family of transcription factors often co-occupy the same regulatory elements and coordinately regulate gene expression with the TAL1 complex [39]. Using ChIP-seq, we identified GATA and ETS motifs at both TAL1 isoforms' binding locations, whereas RUNX1 and E-box were only identified for TAL1-short binding sites (Fig 4D–4H). While this result can help us reveal some of the complexity of the isoforms' function, we need to take into account that our discovery rate is low as a result of the lower enrichment of TAL1-long. The low discovery rate and complex regulation prevent us from pointing at the exact role of each TAL1 isoform as a transcription factor.

Our previous work has demonstrated the connection between enhancer activity and alternative splicing [48]. The enhancer was demonstrated to promote transcription elongation kinetics to promote exon inclusion. Here, we show that TAL1 enhancers regulate its exon 3 alternative splicing. Specifically, deleting the −8 MuTE enhancer promotes exclusion of TAL1 exon 3, while activating TAL1 −60 enhancer via deletion of a CTCF site or tethering of dCas9-p300 has the opposite result (Fig 2B and 2D). This result indicating enhancer activation with exon exclusion is different than the one in our previous work [48]. While we could see increased acetylation at TAL1 exon 3 upon enhancer activation (Fig 2G), we could not find an effect of RNAPII elongation rate on TAL1 alternative splicing (S2K–S2M Fig). TAL1 −8 MuTE enhancer is considered to be a strong/super enhancer [14]. This type of enhancer changes the chromatin not only at the gene's promoter but also at its intragenic region (Fig 1D). Our findings suggest that the broad chromatin domain marked by H3K4me3 and H3K27ac plays a crucial role in regulating both alternative splicing and gene expression. Specifically, our results highlight KMT2B, which is a component of the SET1/COMPASS complexes responsible for the broad methylation of H3K4, as a regulator of alternative splicing of the *TAL1* gene (Fig 2H). Thus, the close proximity between TAL1 promoter and exon 3 (2.5 Kb), as well as the fact that promoter 5 is located in exon 3 (Fig 1A), forms open chromatin at exon 3 promoting inclusion of exon 3 giving rise to TAL1-long.

Overexpressing TAL1-long in transgenic mice promotes T-ALL in only 30% of animals after a latent period of more than 14 weeks [37]. In line with these results, our work did not lead to T-ALL development after overexpression of either isoform. TAL1 expression is reduced in T-cell development to allow activation of E-proteins. Hence, overexpression of both isoforms in mice resulted in a lower number of lymphocytes relative to control (Fig 5C). While lymphocyte development was affected in the same way by both isoforms, TAL1-short leads to hematopoietic stem cells exhaustion while no change was observed in TAL1-long expressing cells. This is a very interesting result that could have been explained mechanistically by monitoring transcription regulation by the isoforms in those cells. Unfortunately, at 14 weeks, the small number of TAL1-short overexpressing cells did not allow us to conduct RNA-seq to further investigate this isoform's targets.

TAL1 harbors 4 active promoters in Jurkat cells and, thus, 4 distinct 5′ UTRs. Our results point at a complex regulation of these elements orchestrated by its enhancers. In addition, our results show that TAL1-short expression is a cause of Jurkat and K562 cell death while TAL1-long expression does not affect cell survival (Figs 4K, 4L, 6C, and 6D). Furthermore, our results indicate that TAL1-short is a stronger erythropoiesis factor, suggesting that the isoforms have differential roles in hematopoiesis. These results demonstrate the importance of a fine-tuned regulation of TAL1 mRNA processing, which is critical for hematopoiesis and cell survival.

## Material and methods

### Ethics statement

Mice were maintained and bred under specific pathogen-free conditions in the Hebrew University animal facility (IACUC: Authority For Biological and Biomedical Models, NIH approval number: OPRR-A01-5011) according to Institutional Animal Care and Use Committee regulations. Mice experiments were approved by the Hebrew University Ethics committee (approval number MD-20-16296-5). At the ethical end point, the mice underwent anesthesia with overdose of ketamine/xylazine mixture and were killed by cervical dislocation.

### Cell lines and plasmids

HEK293T (ATCC Number: CRL-3216) cells were grown in Dulbecco's Modified Eagle's Medium (DMEM) supplemented with 10% fetal bovine serum (FBS). Jurkat (Clone E6-1), MOLT4, CutLL1, CCRF-CEM, K562 cells (ATCC Number: CCL-243), Jurkat NLO, and Jurkat Del-12 were grown in Roswell Park Memorial Institute's medium (RPMI-1640) supplemented with 10% FBS. HEK293T and HEK293T ΔCTCF were grown in DMEM supplemented with 10% FBS. Cell lines were maintained at 37°C and 5% $CO_2$ atmosphere. Cells were transfected with TransIT-X2 transfection reagent (Mirus) following the manufacturer's instructions. After 30 h in culture, plasmid transfected cells were used for experimentation. Silencing KMT2B in Jurkat cells was conducted by transfecting the cells with 20 nM of esi KMT2B (Sigma Cat No. EHU076961) using TransIT-X2 transfection reagent (Mirus) for the duration of 72 h.

TAL1 promoters 1 to 3, 4, 5, and exon 4 were cloned to pGL4.23 (Promega) upstream to the minimal promoter. TAL1 alternative 5′ UTRs TSS1, 2, 4, and 5 were cloned after the minimal promoter and before the firefly luciferase initiation codon.

K562 cells were induced into differentiation by incubation with 20 μM hemin and were harvested after the indicated time of induction.

pcDNA-dCas9-p300 Core plasmids (D1399Y; plasmid #61358 and plasmid #61357) were purchased from Addgene. pSPgRNA (Addgene, plasmid #47108) was used as the gRNA plasmid. The oligonucleotides containing the target sequences were hybridized, phosphorylated and cloned into the plasmid using BbsI sites. The target sequences are provided in S1 Table.

### Virus production

FLAG-TAL1-short/long were cloned to MIGR1 vector (Addgene, plasmid#27490). HEK293T cells were cotransfected with the viral backbone vector pCMV-VSVG and pCL-Eco packaging vectors using polyethylenimine (PEI) transfection reagent. Virus-containing supernatants were collected 24, 48, and 72 h posttransfection following their centrifugation and collection at 1,500*g* for 10 min at 4°C. Jurkat and K562 cells were infected with retroviruses by the *spinoculation* method by centrifuging at 800*g* for 2 h followed by 72 h incubation at 37°C.

shRNA sequence targeting TAL1 3′ UTR (CATAACCACTGAAGGGAAT) was cloned to Tet-pLKO-puro vector (Addgene #8453). For lentivirus production, HEK293T cells were cotransfected with the viral backbone vector, pCMV-dR8.2 dvpr and pCL-Eco packaging vectors using PEI transfection reagent as described above. Jurkat and K562 cells were infected with lentiviruses by the *spinoculation* method as described above. Infected K562 and Jurkat cells were selected with puromycin. shRNA expression was induced with 5 μg/mL tetracycline.

### qRT-PCR

RNA was isolated from cells using the GENEzol TriRNA Pure Kit (GeneAid). cDNA synthesis was carried out with the qScript cDNA Synthesis Kit (QuantaBio). qPCR was performed with

the iTaq Universal SYBR Green Supermix (Bio-Rad) on the Bio-Rad iCycler. The comparative Ct method was employed to quantify transcripts, and delta Ct was measured in triplicate. Primers used in this study are provided in S1 Table.

### 5′ RACE

cDNA was synthesized using an RT primer at exon 4 of the TAL1 gene, containing an Illumina adaptor at the 5′ end (gtctcgtgggctcggagatgtgtataagagacagCATCAGTAATCTCCATCTC). This primer, in conjunction with a template switching oligo (TSO; tcgtcggcagcgtcagatgtgtataa-gagacaGrGrGrG), generates cDNAs containing adaptor sequences at both the 5′ and 3′ ends. The resulting cDNAs are directly amplified using primers for the Illumina adaptor sequences at the cDNA ends.

The libraries and sequencing design led to R1 reads showing the 5′ end of transcripts, and R2 reads with the TAL1 sequence used for targeted amplification. Cutadapt software [49,50], version 2.10, was used to select reads from the expected position in the TAL1 gene, by searching for the sequence **CATCAGTAATCTCCATCTC**ATAGGGGGAAGGTCTCCTC (RACE primer in bold, extending into the TAL1 exon) at the beginning of R2 reads. Low-quality and adapter sequences were then removed, using cutadapt and fastq_quality_filter (FASTX package, v0.0.14), then processed reads were aligned to the genome. Alignment was done with TopHat, version 2.1.1, allowing for 2 mismatches. The genome version was GRCh38 (human), with annotations from Ensembl release 99. The right most position of aligned reads, as TAL1 is on the reverse strand, was used to count abundance at putative TSSs.

### Immunoblotting

For immunoblotting, cells were harvested and lysed with RIPA lysis buffer, and 20 μg/μl of the extracts were run on a 10% TGX Stain-free FastCast gel (Bio-Rad) and transferred onto a nitrocellulose membrane. Antibodies used for immunoblotting were anti-TAL1 (MBS4381279), anti-TCF3 (CST #12258), anti-TCF12 (CST #11825), anti-Cas9 [Danyel Biotech #632628 (Clone TG8C1)], and anti-α-Tubulin (CST #3873).

### Immunoprecipitation (IP)

Cells were washed twice with ice-cold PBS, harvested and lysed for 30 min on ice in 0.5% NP40, 150 mM NaCl, 50 mM Tris (pH 7.5), and 2 mM MgCl2 supplemented with protease inhibitor and RNAse inhibitor (Hylabs). Cell supernatants were collected after centrifugation at 16,000g for 20 min. Anti-FLAG M2 Magnetic Beads (Sigma) were added for 2 h at 4°C. Beads were washed 4 times, boiled in sample buffer and loaded onto a 10% TGX Stain-free FastCast gel, and transferred onto a nitrocellulose membrane, and enhanced chemiluminescence using Clarity Western ECL substrate (Bio-Rad) was used to visualize the protein.

### Luciferase reporter gene assay

HEK293T cells were transfected using TransIT-X2 transfection reagent, according to the manufacturer's protocol. Cells ($15 \times 10^4$ cells/well) were seeded in 96-well plates 24 h before transfection. pGL4.23 plasmid including pRL renilla luciferase reporter vector as an internal control, were mixed in OptiMEM reduced serum medium (Gibco), and incubated with transfection reagent for 20 min at room temperature. Transfection complexes were then gently added into individual wells of the 96-well plate. Cells were harvested using Promega Dual-Glo Luciferase Assay System according to the manufacturer's protocol, 30 h after transfection. The relative light units (RLUs) were normalized for transfection efficiency using the ratio between

firefly and renilla luciferase activity and the fold induction over unstimulated controls was calculated. Each assay was performed in triplicate.

## ChIP

Approximately $9 \times 10^6$ cells per sample were cross-linked for 10 min in 1% formaldehyde at RT and quenched with 0.1 M glycine. Cells were washed twice with cold PBS and lysed with lysis buffer (0.5% sodium dodecyl sulfate [SDS], 10 mM ethylenediaminetetraacetic acid [EDTA], 50 mM Tris–HCl (pH 8), and 1×protease inhibitor cocktail). DNA was sonicated in an ultrasonic bath (Diagenode, Bioruptor pico) to an average length of 200 to 250 bp. Supernatants were immunoprecipitated overnight with 40 μL of precoated anti-IgG magnetic beads (goat anti-rabbit IgG magnetic beads, NEB) previously incubated with the antibody of interest for 4 h at 4˚C. The antibodies used were rabbit anti-H3Ac, acetyl K9 + K14 + K18 + K23 + K27 (Abcam, cat. no. ab47915), and rabbit anti-histone H3 (Abcam, cat. no. ab1791). For ChIP-seq, supernatant was incubated with 20 μL Anti-FLAG M2 Magnetic Beads. Beads were washed sequentially for 5 min each in low-salt (20 mM Tris–HCl (pH 8), 150 mM NaCl, 2 mM EDTA, 1% Triton X-100, 0.1% SDS), high-salt (20 mM Tris–HCl (pH 8), 500 mM NaCl, 2 mM EDTA, 1% Triton X-100, 0.1% SDS), and LiCl buffer (10 mM Tris (pH 8.0), 1 mM EDTA, 250 mM LiCl, 1% NP-40, 1% Nadeoxycholate) and in Tris-EDTA buffer. Beads were eluted in 1% SDS and 100 mM NaHCO3 buffer for 15 min at 65˚C and cross-linking was reversed for 6 h after addition of NaCl to a final concentration of 200 mM and sequentially treated with 20 μg proteinase K. DNA was extracted using magnetic beads (Beckman Coulter, Agencourt AMPure XP, cat. no. A63881). Immunoprecipitated DNA (2 out of 50 μL) and serial dilutions of the 10% input DNA (1:5, 1:2.5, 1:1.25, and 1:0.625) were analyzed by SYBR-Green real-time qPCR. ChIP-qPCR data were analyzed relative to input to include normalization for both background levels and the amount of input chromatin to be used in ChIP. The primer sequences used are listed in S1 Table.

## ChIP analysis

ChIP data Fig 1D: CutLL1: H3K27ac [51] GSM1252938, H3K4me3 [52] GSM732911. MOLT4 and CCRF-CEM: H3K27ac [53] GSM2037790, H3K4me3 [54] GSM4703778. Jurkat: H3K27ac [16] GSM1697882, H3K4me3 [55] GSM945267. K562: H3K27ac and H3K4me3 [56] GSM733656.

TAL1 isoforms binding sites were identified from 2 biological replicates using HOMER (v4.11) [57]. Peak calling was performed with empty-GFP as background, using the "factor" mode and fold change ≥2. Otherwise, default parameters were used. Total TAL1 ChIP data were obtained from GEO database: GSE29180 [39] and analyzed using the same pipeline and parameters. Peak annotation and visualization were done using HOMER annotatePeaks tool.

## RNA sequencing and analysis

Total RNA samples for 3 biological replicates (approximately $1 \times 10^6$ cells) were subjected to sequencing.

Raw reads were processed for quality trimming and adaptors removal using fastx_toolkit v0.0.14 and cutadapt v2.10. The processed reads were aligned to the human transcriptome and genome version GRCh38 with annotations from Ensembl release 106 using TopHat v2.1.1 [49]. Counts per gene quantification was done with htseq-count v2.01 [58]. Normalization and differential expression analysis were done with the DESeq2 package v 1.36.0 [59]. Pair-wise comparisons were tested with default parameters (Wald test), without applying the independent filtering algorithm. Significance threshold was taken as padj < 0.1. In addition, significant

DE genes were further filtered by the log2FoldChange value. This filtering was baseMean-dependent and required a baseMean above 5 and an absolute log2FoldChange higher than 5/sqrt(baseMean) + 0.3 (for highly expressed genes this means a requirement for a fold-change of at least 1.2, while genes with a very low expression would need a 5.8-fold change to pass the filtering).

## Cell proliferation

Suspensions of Jurkat or K562 cells in logarithmic growth at a cell density of $1.6 \times 10^5$/ml were seeded in 12-well plates. Cell cultures were seeded in triplicates and were incubated for the time indicated. Cells were counted post trypan blue staining, and the mean value was calculated for each time point relative to day 1.

## CPT treatment

To impede the dynamics of transcribing RNAPII, cells were treated with CPT (Sigma) to a final concentration of 6 μM for 6 h.

## Mice

C57BL/6J (wild-type) and C57BL/6.SJL (PtprcaPep3b; Ly5.1) (CD45.1) mice were from The Jackson Laboratory.

## Bone marrow transplantation

Donor mice were injected IP with 150 mg/kg 5-fluorouracil (5-FU) 5 days before cell harvesting. Bone marrow cells were extracted as previously described [60]. Cells were then transduced with retroviruses expressing FLAG-TAL1-short/GFP or FLAG-TAL1-long/dt-tomato and incubated for 2 days at 37°C. Recipient mice were irradiated (600 rad) using an XRAD-225 machine, 24 h before transplantation. The irradiated recipients were then divided into 2 groups that were injected with $2.5 \times 10^6$ bone marrow cells by tail vein injection. Mice were maintained for 4 weeks post-irradiation on water containing neomycin.

## Retrovirus generation and bone marrow infection

Virus-containing supernatants were collected 24, 48, and 72 h posttransfection following their concentration in 70,000 x $g$ at 4°C using an ultracentrifuge. Infected bone marrow cells were incubated with concentrated viral supernatants for 4 h in bacterial tubes and then transferred back to 6-well plates for additional 48 h.

## Mixed bone marrow chimera transplantation

Donor mice (C57BL/6.SJL (PtprcaPep3b; Ly5.1) (CD45.1) were injected IP with 150 mg/kg 5-FU 5 days before bone marrow cells harvesting. The bone marrow cells were extracted as previously described [61]. Cells were then transduced with retroviruses expressing TAL1-short-GFP or TAL1-long-dtTomato and incubated for 48 h at 37°C. Recipient C57BL/6J (wild-type) mice were irradiated (850 rad) using an XRAD-225 machine, 24 h before transplantation. The irradiated recipients were injected with equal amounts of bone marrow cells of the 2 transductions by tail vein injection. Mice were maintained for 3 weeks post-irradiation on water containing neomycin. Mice were monitored by blood tests every 3 to 4 weeks for 13 weeks after the bone marrow transplantation.

### Cytokines

The following cytokines were used for bone marrow culture: IL3, IL6 (20 ng/ml), stem cell factor (50 ng/ml), and TPO (100 ng/ml) (Peprotech).

### Flow cytometry antibodies

The following antibodies were used: CD90.2 (Thy 1.2) (53–2.1), CD45.1 (A20), LY6C (HK1.4), LY6G(1A8), CD71 (RI7217), Ter119 (TER-119) (BioLegend).

### Flow cytometry detection of hematopoietic populations

Blood was collected from recipient mice every 3 to 4 weeks post-bone marrow transplantation. Only for B, T cells, and granulocytes, red blood cells were removed from the samples by erythrocytes lysis buffer containing $NH_4Cl$, $KHCO_3$, and EDTA (Sigma). The remaining cells were then resuspended in the relevant antibody mixture diluted in PBS containing 2% FBS. Cells were then analyzed using a CytoFLEX Flow Cytometer (Beckman Coulter). Analysis was performed by FCS express V.6 (De Novo Software). After 14 weeks, the mice were killed and bone marrow and spleen were harvested, stained for the relevant antibodies, and analyzed in flow cytometry. All FCS file were uploaded to http://flowrepository.org/, ID: FR-FCM-Z69L.

## Supporting information

**S1 Fig.** (A and B) RNA was extracted from the indicated cell lines and analyzed by real-time PCR for total mRNA amount of TAL1 relative to CycloA and hTBP reference genes (S1 Data) (A) and for ΔEx3 relative to endogenous TAL1 total mRNA amount. PSI was calculated by ΔEx3 relative to endogenous TAL1 total mRNA (S1 Data) (B). (C) ChIP-seq tracks for H3K27ac and H3K4me3 at the TAL1 locus in the indicated cell-lines (genome build hg19). (PPTX)

**S2 Fig.** (A) Sequencing of 5′ RACE PCR in Jurkat cells aligned to TAL1 locus. Peaks mark the sequence of the 5′ UTR. Red asterisk marks a peak that did not align to any known TSS. (B) Schematic representation of TAL1 mRNA isoforms. Rectangles: exons, black lines: introns; arrow: transcription initiation site. (C and D) Whole cell lysate was extracted from Jurkat NLO cells, expressing TAL1 exogenously, and enhancer mutated cells Jurkat Del-12, and subjected to western blot analysis using the indicated antibodies (S1 Raw Images) (C). RNA was extracted and real-time PCR was conducted TAL1 promoter relative to to CycloA and hTBP reference genes (S1 Data) (D). (E) Schematic representation of TAL1 genomic locus. Rectangles: exons, black lines: introns, arrow: transcription initiation site. (F-H) Whole cell lysate was extracted from HEK293T cells and CTCF mutated cells, HEK293T ΔCTCF, and subjected to western blot analysis using the indicated antibodies (S1 Raw Images) (F). RNA was extracted and analyzed by real-time PCR for each of TAL1's promoters relative to CycloA and hTBP reference genes (S1 Data) (G). (H) HEK293T cells were transfected with either dCas9-p300 core (mut) or dCas9-p300 core (WT) with 4 gRNAs targeted to the TAL1 −60 enhancer for 30 h. Whole cell lysate was extracted and subjected to western blot analysis using the indicated antibodies (S1 Raw Images). (I and J) Jurkat cells were transfected with siKMT2B or a negative control siRNA (siGFP), and RNA was extracted 72 h posttransfection. Real-time PCR was performed to KMT2B total mRNA amount relative to CycloA and hTBP reference genes (S1 Data) (I) and total mRNA amount of TAL1 relative to CycloA and hTBP reference genes (S1 Data) (J). The mean was calculated from 3 independent biological experiments, each performed with 3 technical replicates. (K-M) Jurkat cells were treated with 6 μM CPT for 6 h and analyzed by real-time PCR for total mRNA amount of CycloA relative to 18S reference gene transcribed by

Polymerase I (K) and TAL1 relative to 18S reference gene (L) and for ΔEx3 relative to TAL1 total mRNA amount. PSI was calculated by ΔEx3 relative to TAL1 total mRNA (S1 Data) (M).
(PPTX)

**S3 Fig.** (A-C) HEK293T cells were cotransfected with TAL1 promoters: promoters 1–3, 4, 5, and exon 4 as a negative control. The second plasmid was an empty vector, TAL1-short, or TAL1-long. After 30 h, RNA was extracted and real-time PCR was performed to TAL1 total mRNA amount relative to CycloA and hTBP reference genes (S1 Data) (A) and whole cell lysate was extracted and subjected to western blot analysis using the indicated antibodies (S1 Raw Images) (B) and RNA was extracted and analyzed by real-time PCR for luciferase mRNA relative to renilla mRNA (S1 Data) (C).
(PPTX)

**S4 Fig.** (A) Jurkat cells were infected with either empty vector, GFP-TAL1-short, or GFP-TAL1-long. GFP was immunoprecipitated and interacting proteins were detected with indicated antibodies (S1 Raw Images). (B-J) Jurkat cells were infected with either empty vector, FLAG-TAL1-short, or FLAG-TAL1-long. ChIP-seq was performed with anti-FLAG magnetic beads. In addition, we analyzed available data for TAL1-total (see methods). Whole cell lysate was extracted and subjected to western blot analysis using the indicated antibodies (S1 Raw Images) (B). ChIP-seq tracks are shown at the indicated locus for each biological experiment. Arrows indicate known TAL1-total binding site (C). ChIP-seq average signal for TAL1-total, TAL1-short and TAL1-long, empty vector and input as a function of distance from TAL1-total peaks (S1 Data) (D). ChIP-seq peaks across genomic regions (S1 Data) (E). Most abundant DNA sequence motifs identified in ChIP-seq peaks (F-J). (K-R) Jurkat cells were infected with shRNA to TAL1 3′ UTR and later with empty vector, FLAG-TAL1-short or FLAG-TAL1-long. Induction with tetracycline was conducted for 72 h, and silencing was measured using real-time PCR for total mRNA amount of endogenous TAL1 relative to CycloA and hTBP reference genes (S1 Data) (K). RNA-seq was preformed on 3 biological replicates and analyzed for isoforms specific targets. In addition, we analyzed available data for TAL1-total (see methods). Venn diagrams showing numbers of distinct and common RNA-seq targets and peak-associated genes of TAL1-total (L) and TAL1-short (M). Real-time PCR for total mRNA amount of 3 TAL1-short targets was performed relative to CycloA and hTBP reference genes (S1 Data) (N). Bar chart of top 4 enriched terms from the GO_Biological_Process_2018 gene set library for TAL1-total (O) and TAL1-short (P) and TAL1-long (Q). Heatmap and statistics for expression level of apoptosis pathway genes from GSEA plot (R).
(PPTX)

**S5 Fig.** (A-K) Equal numbers of bone marrow cells from 5FU-treated CD45.1 wild-type mice were transduced with retroviruses expressing either TAL1-short-GFP or with TAL1-long-dtTomato. A mixture of 1:1 ratio of the transduced bone marrow cells was then transplanted into lethally irradiated CD45.2 wild-type recipient mice. Mice were killed 14 weeks after BMT, and the bone marrow cells were harvested and analyzed using flow cytometric analysis. (A) Representative flow cytometry dot plots of Ly6C vs. Ly6G staining gated on CD11b-positive cells that are GFP−/dtTomato− (untransduced, left panel), dtTomato+ (Tal1-long, middle panel) or GFP+ (Tal1-short, right panel) splenocytes. Regions represent the following populations: R1− CD11b+, Ly6C−, Ly6G− (circulating monocytes), R2− CD11b+, Ly6Chi, Ly6G− (inflammatory monocytes), R3− CD11b+, Ly6Cint, Ly6G+ (granulocytes). (B) Bar graph summarizing results in c (S1 Data). (C) Representative flow cytometry dot plots analysis of CD71 and Ter119 staining gated on CD45-negative cells that are GFP−/dtTomato− (untransduced, left panel), dtTomato+ (Tal1-long, middle panel), or GFP+ (Tal1-short, right panel). Regions

S0-S5 represent the stage in erythrocyte differentiation. (D) Bar graph summarizing results in e (S1 Data). Left panel: percentages of CD71− Ter119− cells (the S0 phase in erythrocytes differentiation); right panel: CD71+ Ter119hi (the S3 phase in erythrocytes differentiation. (E) Frequencies of the subpopulation of erythrocytes in the different differentiation stages based on the CD71 and Ter119 expression (S1 Data). (F) Representative flow cytometry dot plots of Thy1.2 vs. CD19 staining gated on GFP−/dtTomato− (untransduced, left panel), dtTomato+ (Tal1-long, middle panel), or GFP+ (Tal1-short, right panel) bone marrow cells. (G) Bar graph summarizing results in a (S1 Data). (H) Representative flow cytometry histograms of CD11b staining gated on GFP−/dtTomato− (untransduced, left panel), dtTomato+ (Tal1-long, middle panel), or GFP+ (Tal1-short, right panel) bone marrow cells. (I) Bar graph summarizing results in c (S1 Data). (j) Representative flow cytometry dot plots of Ly6C vs. Ly6G staining gated on CD11b-positive cells that are GFP−/dtTomato− (untransduced, left panel), dtTomato + (Tal1-long, middle panel), or GFP+ (Tal1-short, right panel) splenocytes. Regions represent the following populations: R1− CD11b+, Ly6C−, Ly6G− (circulating monocytes), R2− CD11b +, Ly6Chi, Ly6G− (inflammatory monocytes), R3− CD11b+, Ly6Cint, Ly6G+ (granulocytes). (K) Bar graph summarizing results in e (S1 Data). Two-tailed paired $t$ test, $^*<0.05$, $^{**}<0.01$, $^{***}<0.001$, $^{****}<0.0001$, sd. ($n = 6$).
(PPTX)

**S6 Fig.** (A-I) K562 cells were infected with inducible shRNA against the 3′ UTR of TAL1. In addition, the cells were infected with MIGR1 plasmid with GFP at the C-terminal followed by an IRES element. Infection was with either empty vector, TAL1-short, or TAL1-long. Following induction with tetracycline for 72 h, cells were treated with 20 uM hemin to promote erythroid differentiation for the indicated time points. RNA was extracted silencing was measured by real-time PCR for total endogenous TAL1 relative to CycloA and hTBP reference genes (S1 Data) (A) and whole cell lysate was extracted and subjected to western blot analysis using the indicated antibodies (S1 Raw Images) (B and C). RNA was analyzed by real-time PCR for total mRNA amount of α-, β-, and γ-globin and SLCA4 relative to CycloA and hTBP reference genes after 72 h (S1 Data) (D) and 96 h (S1 Data) (E) and following treatment with hemin (inducing differentiation) after 72 h (S1 Data) (F) and 96 h (S1 Data) (G). Cells were seeded and counted every day following labelling by trypan blue following treatment with hemin (inducing differentiation). Live cells are plotted in H and trypan blue-labeled cells are plotted in I (S1 Data) (the corresponding plots for cells without hemin treatment can be seen in Fig 6C and 6D). Plots represent the mean of 3 independent experiments ± SD ($^*<0.05$, $^{**}P < 0.01$, $^{***}P < 0.001$, and $^{****}P < 0.0001$, Student $t$ test).
(PPTX)

**S1 Raw Images. All raw data used for figures.**
(PDF)

**S1 Table. Primers used in this study.**
(PDF)

**S1 Data. Raw data used to compile the graphs in this study.**
(XLSX)

## Acknowledgments

We thank Roi Gazit and Ido Goldstein for their expert advice throughout this work. We thank Thomas Look for Jurkat NLO and Jurkat Del-12 cells. We thank Richard A. Young for HEK293T parental and HEK293T ΔCTCF cells.

## Author Contributions

**Conceptualization:** Aveksha Sharma, Shani Mistriel-Zerbib, Klil Cohen, Yotam Drier, Michael Berger, Maayan Salton.

**Formal analysis:** Eden Engal, Mercedes Bentata, Yuval Nevo, Inbar Plaschkes, Yotam Drier.

**Investigation:** Aveksha Sharma, Shani Mistriel-Zerbib, Rauf Ahmad Najar, Eden Engal, Mercedes Bentata, Nadeen Taqatqa, Sara Dahan, Klil Cohen, Shiri Jaffe-Herman, Ophir Geminder, Mai Baker, Gillian Kay, Michael Berger, Maayan Salton.

**Methodology:** Aveksha Sharma, Shani Mistriel-Zerbib, Rauf Ahmad Najar, Michael Berger, Maayan Salton.

**Project administration:** Gillian Kay.

**Supervision:** Maayan Salton.

**Writing – original draft:** Maayan Salton.

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
