## [Editor Report · Decision Letter 0]

25 Apr 2023

Dear Dr Salton, 

Thank you for submitting your manuscript from Review Commons entitled "Hematopoiesis and cell growth are differentially regulated by TAL1 isoforms" for consideration as a Research Article by PLOS Biology.

Please accept my sincere apologies for the great delay in getting back to you as we consulted with an academic editor with the relevant expertise about your submission. I'm writing to let you know that we would like to send a minor revision based on your responses to the previous reviews at Review Commons.

However, before we can invite a revision, we need you to complete your submission by providing the metadata that is required for full assessment. To this end, please login to Editorial Manager where you will find the paper in the 'Submissions Needing Revisions' folder on your homepage. Please click 'Revise Submission' from the Action Links and complete all additional questions in the submission questionnaire.

Once your full submission is complete, your paper will undergo a series of checks in preparation to issue the decision. After your manuscript has passed the checks we will be able to issue the revision. To provide the metadata for your submission, please Login to Editorial Manager (https://www.editorialmanager.com/pbiology) within two working days, i.e. by Apr 27 2023 11:59PM.

Kind regards,

Richard

Richard Hodge, PhD

Associate Editor, PLOS Biology

rhodge@plos.org

PLOS

---

## [Editor Report · Decision Letter 1]

27 Apr 2023

Dear Dr Salton,

Thank you for your patience while we considered your revised manuscript from Review Commons entitled "Hematopoiesis and cell growth are differentially regulated by TAL1 isoforms" for publication as a Research Article at PLOS Biology. 

Based on our Academic Editor's assessment of your responses to the previous reviews at Review Commons and the revision, I am pleased to say that we are likely to accept this manuscript for publication, provided you satisfactorily address the following data and other policy-related requests that I have provided below (A-I):

(A) We would like to suggest the following modification to the title:

“"Isoforms of the TAL1 transcription factor have different roles in hematopoiesis and cell growth"

(B) In the Methods section of the manuscript, please include the full name of the IACUC/ethics committee that reviewed and approved the animal care and use protocol/permit/project license. Please also include the specific approval number issued by your IACUC to conduct the study, as well as the method of euthanasia used to sacrifice the mice. 

(C) You may be aware of the PLOS Data Policy, which requires that all data be made available without restriction: http://journals.plos.org/plosbiology/s/data-availability. For more information, please also see this editorial: http://dx.doi.org/10.1371/journal.pbio.1001797

-Supplementary files (e.g., excel). Please ensure that all data files are uploaded as 'Supporting Information' and are invariably referred to (in the manuscript, figure legends, and the Description field when uploading your files) using the following format verbatim: S1 Data, S2 Data, etc. Multiple panels of a single or even several figures can be included as multiple sheets in one excel file that is saved using exactly the following convention: S1_Data.xlsx (using an underscore).

-Deposition in a publicly available repository. Please also provide the accession code or a reviewer link so that we may view your data before publication. 

Figure 2A-H, 3A-B, 4C-H, 4J-L, 5C, 5E, 5G, 6A-D, S1A-B, S2D, S2G, S2I, S2J-M, S3A, S3C, S4D-E, S4K, S4N, S5B, S5D-E, S5G, S5I, S5K, S6A, S6D-I

(D) Thank you for depositing the RNA-seq (GSE214833) and ChIP-seq (GSE216684) data in the GEO database. However, I note that the data is currently private and is scheduled to be released on October 1st 2023. At this time, we ask that you please make both datasets publicly available before publication and amend the Data Availability Statement to remove any references to access tokens. 

(E) Please also ensure that each of the relevant figure legends in your manuscript include information on *WHERE THE UNDERLYING DATA CAN BE FOUND*, and ensure your supplemental data file/s has a legend.

(F) We require the original, uncropped and minimally adjusted images supporting all blot and gel results reported in the following figures:

Figure 1C, 4A, S2C, S2F, S2H, S3B, S4A-B, S6B-C

We will require these files before a manuscript can be accepted so please prepare and upload them now. Please carefully read our guidelines for how to prepare and upload this data: https://journals.plos.org/plosbiology/s/figures#loc-blot-and-gel-reporting-requirements

(G) For figures containing FACS data (Figure 5B, 5D, 5F, S5A, S5C, S5F, S5H, S5J), please provide the FCS files and a picture showing the successive plots and gates that were applied to the FCS files to generate the figure. We ask that you please deposit this data in the FlowRepository (https://flowrepository.org/) and provide the accession number/URL of the deposition in the Data Availability Statement in the online submission form.

(H) Please note that per journal policy, the model system/species studied should be clearly stated in the abstract of your manuscript.

(I) Please ensure that your Data Statement in the submission system accurately describes where your data can be found and is in final format, as it will be published as written there. 

We expect to receive your revised manuscript within two weeks. 

*Published Peer Review History*

*Press*

Kind regards,

Richard

Richard Hodge, PhD

Associate Editor, PLOS Biology

rhodge@plos.org

PLOS

---

## [Editor Report · Decision Letter 2]

30 May 2023

Dear Dr Salton,

Thank you for the submission of your revised Research Article "Isoforms of the TAL1 transcription factor have different roles in hematopoiesis and cell growth" for publication in PLOS Biology. On behalf of my colleagues and the Academic Editor, Connie Eaves, I am pleased to say that we can accept your manuscript for publication, provided you address any remaining formatting and reporting issues. These will be detailed in an email you should receive within 2-3 business days from our colleagues in the journal operations team; no action is required from you until then. Please note that we will not be able to formally accept your manuscript and schedule it for publication until you have completed any requested changes.

PRESS

Best wishes, 

Richard

Richard Hodge, PhD

Associate Editor, PLOS Biology

rhodge@plos.org

PLOS
